# Assessment of the Psychophysical Sphere and Functional Status of Women Aged 75–90 Living Alone and in Nursing Homes

**DOI:** 10.3390/ijerph18179028

**Published:** 2021-08-27

**Authors:** Justyna Traczyk, Agnieszka Dębiec-Bąk, Anna Skrzek, Małgorzata Stefańska

**Affiliations:** 1ECURA Hjemmetjenester og BPA AS, Østensjøveien 14, 0661 Oslo, Norway; j.traczyk89@gmail.com; 2Faculty of Physiotherapy, Wroclaw University of Health and Sport Sciences, 35 Paderewskiego Street, 51-612 Wrocław, Poland; agnieszka.debiec-bak@awf.wroc.pl (A.D.-B.); anna.skrzek@awf.wroc.pl (A.S.)

**Keywords:** old age, quality of life, functional status, physical activity, depression, gerontology

## Abstract

**Aim:** To assess the psychophysical sphere and functional status of women aged 75–90 living alone and in nursing homes. **Methods:** 23 women living in nursing homes (NH) and 20 living alone (HOME) underwent the following tests: Up and Go Test, Chair Stand Test, assessment of daily physical activity levels using pedometers, Mini-Mental State Examination, Groningen Activity Restriction Scale, Geriatric Depression Scale, and WHOQOL-Bref. **Results:** It was shown that the subjects living by themselves performed a greater amount of daily physical activity, although there was not significant difference between the two groups in the Up and Go Test result. There were not statistically significant differences in the self-assessment of the mental sphere, but significant differences were found in the self-evaluation of the physical sphere. In the NH group, subjects with and without depression did not differ in terms of the amount of daily physical activity and functional test results. Women with depression from the HOME group were less physically active and had worse functional fitness. **Conclusions:** Women living alone performed a greater amount of daily physical activity, but the functional status of women in both groups did not differ in a statistically significant way. The groups didn’t differ statistically significantly in terms of psychological self-assessment.

## 1. Introduction

The aging process is an irreversible physiological process that is becoming a characteristic demographic phenomenon in contemporary society. On the one hand, reaching an advanced age brings a sense of satisfaction, fulfilment, and understanding of the meaning of one’s life but, on the other hand, it can become a source of physical and mental pain leading to dissatisfaction or depression. Involutional changes caused by aging limit the ability to live an independent life, decrease physical activity, and undermine the sense of security and the ability to make independent decisions. The speed of aging of the body in the so-called “autumn years” may be influenced by various factors—genetic, physiological, psychological, but also social [1].

Elderly people are often unable to fully take care of all their needs, which is the reason why they are often forced to live with caregivers, in nursing or retirement homes [2,3].

Current research confirms that physical activity affects the mental condition and fitness of the elderly. Regular physical exercise affects the sense of mental well-being and slows down involution changes in the body [4,5,6]. Unfortunately, among the elderly, a reduction in daily activity is observed. Often the reason for the decrease in the level of activity is health deficits and strain on the musculoskeletal system, which in turn affects the faster need for help from third parties or care institutions in further existence [7,8]. Assessment of physical abilities, matching appropriate forms of activity and, above all, the promotion of an active lifestyle seem essential and necessary among the elderly [9,10,11].

Depression and stress connected with changing the place of residence may reduce functional fitness and negatively affect overall quality of life [12,13]. Ongoing research on the quality of life of senior citizens makes it possible to better understand and control socio-medical and care-giving issues. It also leads to more efficient treatment, care, and rehabilitation [14].

Knowing the determinants of life satisfaction in old age can contribute to our improved understanding of the needs, problems, and priorities of old people. It also translates into better organization of personalized care and assistance. Observation and learning about factors that determine the physical and mental well-being of the elderly depending on their place of residence can be an important source of information in planning various treatment and rehabilitation programs. Such knowledge may confirm the need for a holistic approach to elderly people that would account for their most important real needs and expectations [15,16].

Therefore, the aim of this study was to compare the psychophysical condition and functional status of elderly women living alone or in nursing homes, taking into account the integration of physical activity and the mental sphere in everyday life, and to review the evidence regarding the choice of therapeutic and preventive actions [17,18].

## 2. Material and Methods

### 2.1. Study Group

The study group consisted of 43 women selected on the basis of the adopted criteria out of 100 volunteers who applied for the study. The inclusion criteria for the study were: age between 75 and 90 years (the so-called old age according to WHO [19]), no contraindications in the cardiovascular and musculoskeletal systems for performing the functional tests, scoring a minimum of 24 points in the mental state assessment according to the MMSE scale (Mini-Mental State Examination), which demonstrates the capacity to maintain logical verbal contact and the mental ability necessary to consciously fill in the questionnaires [20], and sufficiently good physical fitness and general health to perform basic daily activities (no need for regular, long-term care from third parties).

All the qualified persons received detailed information about the purpose and method of the research and gave their written consent to participate. The project received a positive opinion in accordance with the Resolution of the Senate Committee on Ethics of Scientific Research of the University School of Physical Education in Wroclaw (Poland) of 25 May 2014.

### 2.2. Research Methods

The research was conducted in the period from May to September 2019 in the morning hours (9:00–11:00 a.m.). In order to reduce physical and mental fatigue, tests were performed immediately after the completion of morning hygiene activities and after breakfast.

The subjects were assessed by estimating their daily physical activity and functional fitness using the Up and Go and Chair Stand functional tests. The subjects’ daily physical activity was recorded by means of pedometers, which for seven consecutive days measured the number of steps/day n, distance/day km, consumed energy cal, and step length cm. After that, mean values were calculated for each study participant. Functional fitness was assessed using the Up and Go Test, which measured the time needed to get up from the chair, walk a distance of 3 m, turn, return to the chair and sit down. The second test from this group was the Chair Stand Test, which assessed the strength of lower-extremity muscles. The test was based on determining the number of alternating chair-stands and sit-downs performed within 30 s [21,22,23,24].

The subjects’ mental status was assessed using the Mini-Mental State Examination (MMSE) and the Geriatric Depression Scale (GDS). Subjective functional assessment was performed using the Groningen Activity Restriction Scale (GARS), while the subjects’ quality of life was assessed on the basis of the World Health Organization Quality of Life-Bref (WHOQOL-Bref). Established Polish adaptations were used in this study [25,26,27,28,29].

The Mini-Mental State Examination (MMSE) questionnaire determined cognitive functions by assessing the correctness of responses to 30 tasks regarding orientation in time and space, memory, attention and counting, recall, naming, repetition, understanding, reading, writing, and drawing. Values of 27–30 points were deemed to indicate normal cognitive functioning. A score of 24–26 points indicates a cognitive impairment without dementia. A score lower than 24 points was a criterion for excluding a person from the study [29,30].

The Geriatric Depression Scale (GDS) is a screening test for depression symptoms in the elderly. In the study, the subjects were asked to provide answers to 30 yes–no questions, which were then assessed according to the key. Obtaining 0–9 points was considered a normal result (no depression), 10–19 points indicated mild depression, while a score of ≥20 points indicated severe depression [31].

The Groningen Activity Restriction Scale (GARS) was used to evaluate functional efficiency with regard to performing everyday activities (independent movement, shopping, doing household chores, etc.). Each of the 18 questions in the questionnaire was assessed on a 4-point scale, in which 1 point meant the ability to perform a given activity fully independently, while 4 points meant that the subject was unable to perform a given task at all [32,33].

The study also used the World Health Organization Quality of Life–Bref (WHOQOL–Bref) questionnaire, which is intended for subjective assessment of the quality of life of healthy and disabled people for clinical and cognitive purposes. The questionnaire used in the study contained 26 questions assessing the quality of life in four domains: physical, psychological, social, and environmental. Additionally, it contained two questions about the individual perception of one’s own quality of life and health. Each question was assessed on a scale of 1 to 5 points (the higher the number of points, the higher the quality of life) [34].

### 2.3. Statistical Analysis

The Shapiro–Wilk test was used to check the distribution of quantitative data parameters, which turned out to be close to normal in most cases. The equality of variances was confirmed using the Levene’s test. Descriptive statistics were calculated (mean and SD). The significance of differences between the two groups was checked using the Student’s T-test for independent samples or the Mann–Whitney U test.

The interaction between the place of residence and depression, physical activity and psychophysical self-esteem was performed using a two-way ANOVA. In the event of a statistically significant interaction, the post hoc Scheffe test was performed. In the absence of a normal distribution, the ANOVA Kruskal–Wallis and post hoc multiple comparison mean rank test was used.

Additionally, to determine the quantity of the effect of differences between examined groups, the corrected Cohen’s d test was used. The effect size of interaction was calculated by Eta squared (η^2^) and then transform to Cohen’s d value [35]. Values of the Cohen’s d test ≥ 0.8 proved high strength of the observed effect.

All calculations were made in the Statistica 13.1 program and using the statistical calculators http://www.psychometrica.de/effect_size (access date 15 August 2021).

## 3. Results

The study participants were divided into two groups. The adopted criterion of division was the place of residence. The HOME group consisted of 20 women living independently at home. The NH group included 23 women living in nursing homes in Wrocław (Poland). The study groups did not differ significantly in terms of age and weight (Table 1).

Daily physical activity was recorded for all the subjects, which in the HOME group turned out to be statistically significantly higher than in the NH group. The greatest differences were observed in the values of the parameters describing distance/day km, consumed energy cal and steps/day n. The difference in the results of the Chair Stand Test also turned out to be statistically significant. Significantly more repetitions were recorded in the HOME group compared to the NH group. However, no significant differences between the groups were observed in the results of the Up and Go test (Table 2).

The self-assessment questionnaires showed no statistically significant differences between the groups in terms of cognitive functions (MMSE) and the incidence of depression (GDS). Also, quality of life assessed by means of the WHOQOL–Bref did not differ significantly, with the exception of the physical domain, where differences in the self-assessment of the physical sphere were noted between the groups. In the NH group, quality of life was assessed as worse in the physical domain (WHOQOLD Phys.). Statistically significant differences were also observed in the self-assessment of functional capacity to perform basic everyday activities (GARS). In the NH group, the average value was 8 points higher than in the HOME group, which means that the NH group was less independent (Table 3).

In both study groups, the average GDS score was 6 points, with quartile deviation of 5 and 6.5 points. This means that in both groups there were subjects whose obtained result did not indicate the occurrence of depression (values less than 9 points) and subjects who obtained 10 points and more, which could indicate a mild form of depression. In order to verify whether depressed mood affects physical activity and self-assessment with regard to psychophysical state, two subgroups were distinguished in both study groups. Subgroup 1 consisted of people who scored no more than 9 points in the GDS (Depression NO), whereas Subgroup 2 comprised subjects who scored at least 10 points in the test (Depression YES) (Figure 1).

Taking into account the division into subgroups, no statistically significant difference was observed between the level of physical activity of depressed and non-depressed women living in nursing homes. On the other hand, statistically significantly higher parameters describing the physical activity of the subjects without depression were recorded compared to subjects with depression who lived at home (Table 4). 

When analysing the results of the self-assessment questionnaires, statistically significant differences were observed between the results of subjects with and without depression living in nursing homes. The women without depression obtained a lower average value in the GARS and higher average values in the WHOQOL–Bref, both in total and in each of the analysed domains. A similar relationship was observed in women living in their homes, except for the lack of significant differences in the WHOQOL–Bref scale in the mental and social spheres (Table 4).

The analysis of variance confirmed the significance of differences in the parameters describing the physical activity of people living in homes and in nursing homes. On the other hand, no differences were found between the results of the self-assessment scales used, except for the results of the GARS scale and the WHOQOL physical domain. Women living in houses had a statistically significantly higher daily number of steps, longer distance, more calories consumed, and longer stride length (Table 5).

Regardless of the place of residence, people with confirmed depression on the basis of GDS results were characterized by lower daily physical activity. Depressed and non-depressed people differed significantly in terms of scores on nearly all self-assessment scales except for the Mini-Mental State Examination (MMES) (Table 5).

By examining the joint effect of the place of residence and depression on the results of the tests used, the significance of differences in studies describing physical activity was confirmed. In the study group living in their own home, it was confirmed that people with depression were significantly less physically active than people without depression. Also, significantly lower daily activity was observed in people without depressive symptoms living in nursing homes compared to people without depression living in their own homes. The significance of differences in physical activity between people with and without depression and those living in nursing homes was not confirmed. The differences were mainly concerned the number of steps/day and consumed energy and the length of distance/day (Table 5). The results show that people living alone in their homes are characterized by higher physical activity, regardless of their mental well-being. People with depression living in their homes were characterized by higher daily activity compared to those living in nursing homes without depression.

## 4. Discussion

The aging of societies around the world triggers medical, scientific, and social actions intended to analyse various determinants of this process and search for therapeutic activities that may positively affect the mental and physical well-being of seniors. This would slow down involutional processes and, consequently, extend the possibility of independent functioning of the elderly in society, while limiting the expenditure on institutional care and treatment [36,37].

The advancement of chronological and biological age and the accompanying diseases and disabilities have a negative impact on the condition of old people. The deteriorating physical and mental fitness of seniors causes physical discomfort and decreases psychological well-being. This may impede their ability to independently perform basic daily activities, cause dependence on the help of third parties, contribute to social isolation, and consequently lead to a deterioration of quality of life [38].

The diversity of old people’s experiences and behaviours is reflected by their living conditions, the amount of care they receive, their perception of the world, and self-assessment. This may mean that the quality of life of the elderly affects their health and physical condition [39].

In this study, an attempt was made to determine the influence of the place of residence of elderly women, their expectations, subjective assessment of quality of life, level of everyday physical activity, and functional status. An analysis of the research results showed that although seniors living in their homes performed more physical activity per day, no significant differences were observed between them and residents of nursing homes in terms of the Up and Go Test results, indicating similar functional physical fitness of both groups.

Research by Bujnowska-Fedak et al. [40] on older people living by themselves and in nursing homes, which focused on their functional status in particular, demonstrated that subjects in both groups were characterized by similar efficiency of the musculoskeletal system and general motor coordination assessed on the basis of the Up and Go Test. However, having reviewed the literature on the subject, it seems that the functional status of old people has a strong positive relationship with an active lifestyle and rehabilitation used in institutional care institutions, such as nursing homes [37,41].

In this research, analysis of the studied groups in terms of their psychological self-assessment (MMSE, GDS, WHOQOL–Bref) did not show any statistically significant differences between them. However, significant differences were observed in their self-assessment of the physical domain. The studied women living in nursing homes assessed their ability to perform basic everyday activities (GARS) lower than the group of women living at their homes; similarly, the former assessed their quality of life in the physical domain (WHOQOLD Phys.) as worse than the latter.

Saxen’s [42] research also addressed the subject of public health in the context of physical activity. The potential mental health benefits of physical activity were demonstrated. The research authors also reviewed the documented link between mental disorders and a lack of regular physical activity.

In this research project, a correlation was observed in both groups between the amount of physical activity, the functional tests, and the GARS test results. The self-assessment of less physically active women was worse than that of their more active peers. Moreover, in the HOME group, a correlation was observed between physical activity and the results of the GDS test. The higher the GDS score (which could indicate a pessimistic mood or depression), the lower the recorded level of physical activity.

There exists extensive literature on the subject that demonstrates the benefits of physical activity for mental health. The latest research confirms the positive effect of physical activity on cognitive functions and its beneficial effects in the prevention and treatment of dementia. Stanton et al. [17] conducted a large-scale study demonstrating that long-term participation in physical activity reduces the risk of future depressive disorders. The available evidence suggests that physical activity performed at a regular frequency, intensity, and duration substantially decreases the likelihood of developing future depressive illness, while maintaining cardiorespiratory and muscular fitness by seniors may afford significant benefits in improving their general health.

Depression in old age not only negatively affects social life as well as the course and results of the treatment of somatic diseases, but also contributes to lower quality of life. The study by Pacian et al. [15] was intended to analyse the relationship between the subjective assessment of quality of life and the risk of depression in old people. The research project (comprising 82 subjects) used the WHOQOL–Bref questionnaire to measure the subjects’ quality of life, Yesavage’s Geriatric Depression Scale (GDS) to measure the risk of developing symptoms of depression, and a questionnaire describing demographic and clinical data. Statistically significant relationships were observed, showing that the values of the mental and social domains decrease with age (up to 75 years).

This research also analysed the impact of depression on the results observed in both groups. It was shown that women with depression living in nursing homes did not differ in terms of daily physical activity and functional test results from women without depression in the same group. Subjects with depression who lived in their homes were less physically active and had worse functional capacity compared to women from the same group without depression. However, subjects with depression living in their own homes were characterized by greater physical activity compared to those living in nursing homes, regardless of their depression.

Analyses of depression in old people justify the conclusion that depression is associated with a reduced efficiency of executive functions [18,43,44,45]. On the other hand, Andreas et al. [46] demonstrated in their research that the risk of depression in seniors increases as anxiety states increase. Nowak-Kapusta et al. [47], whose study involved 411 people with symptoms of moderate and severe depression aged 65 and over living in nursing homes in Śląskie Province, emphasized that therapeutic work and appropriate support of institutional care specialists is necessary to minimize the amount of free time spent alone, which negatively affects physical and functional activity. Olsen et al. [48] conducted a study on depression among 58 residents of a nursing home in Norway using the Norwegian version of the Cornell Scale for Depression. Significant depression was observed in old persons with severe dementia, which was associated with reduced functional fitness.

It was also observed in this research that in both groups (NH and HOME), the psychophysical self-assessment of women with depression was lower than of those without depression. In the NH group, differences were found in all self-assessment questionnaires, whereas in the HOME group there was no significant difference in the assessment of quality of life in the mental and social domains.

In a study of seniors living in their homes, in nursing homes and regularly visiting day care centers, Seddigh et al. [49] observed a different gradient of depression symptoms and quality of life assessment depending on the place of residence. Having analysed the subjects’ depression and self-assessment, the authors stressed the importance of social support for enriching the leisure time by providing a variety of cultural, sports, educational and religious programs, which results in a better assessment of the seniors’ mental and social domains.

Puciato et al. [50] and Żurek et al. [51] reached similar conclusions. Having studied a group of 1013 older working-age population, Puciato et al. [50] noticed that the highest average indicators of overall quality of life, as well as subjectively perceived health and quality of life in the physical, mental, social, and environmental spheres, were shown by those subjects whose physical activity was the highest. Observations conducted by Żurek et al. [51] confirmed the existence of a strong positive correlation between the level of physical activity, the level of self-preservation, and the four domains of quality of life.

Developed countries are facing the difficult challenge of providing social care for seniors, who are an inseparable and important element of the social landscape. Never before have they accounted for such a large proportion of society, which is why their quality of life is becoming increasingly important. This article is intended as contribution to the discussion on the aging population and the quality of life of senior citizens. It analyses elements such as health, contacts with family, active participation in social life, mental well-being, opportunities for development, or the possibility of seeing one’s plans through, taking into account older people’s needs, possibilities, and values. The progress of civilization in the field of medicine, coupled with a general improvement of economic and social conditions, not only makes seniors an increasingly more numerous group in society, but also prolongs this stage of life. In developed countries, the tradition of multi-generational homes is disappearing, which is why older people increasingly often live on their own or in institutional care centers. Home dwelling requires the ability to cater to all life needs on one’s own, and thus entails higher levels of physical activity. Care institutions relieve their residents from such duties, which in turn reduces their physical activity, although it is not connected with a worse functional status, as demonstrated by this research. This phenomenon seems to be important because both this study and studies contained in the overview of literature presented above show a high correlation between the mental and physical well-being of the elderly. Therefore, irrespective of their place of residence, seniors should be involved in various types of social programs increasing their activity as far as possible.

The obtained results of the conducted analyses make it possible to attempt to continue the research by extending it to more numerous groups of respondents. This will allow for verifiably broader relationships of the analysed variables, such as the history of life in a disability in the component with physical activity, education, or past profession, as well as having contact with family and friends in relation to mental well-being of seniors.

## 5. Research Limitations

The main limitation of the presented research is the moderate sample size. Continuing the research and increasing the study group involving both women and men will make the observed effects more credible. It will allow the analysis of the relationship between the life history of the elderly, place of residence, and mental well-being. Additionally, the presented research is cross-sectional, because it was impossible to conduct a longitudinal study focusing on the same subjects observed many times over a longer period of time. Research conducted over a long period of time is more likely to provide information needed to understand change mechanisms and other factors influencing human behaviour. 

## 6. Conclusions

Subjects living alone performed a greater amount of daily physical activity and the difference between the two groups was significant. There were no statistically significant differences in the self-assessment of the mental sphere, but significant differences were found in the self-assessment of the physical sphere, which was rated as better by the subjects from the HOME group. Additionally, the self-assessment of less physically active subjects was lower than that of more active ones. In the HOME group, women with depression were less physically active and had worse functional status than their peers without depression. Such differences were not observed in the NH group.

## Figures and Tables

**Figure 1 ijerph-18-09028-f001:**
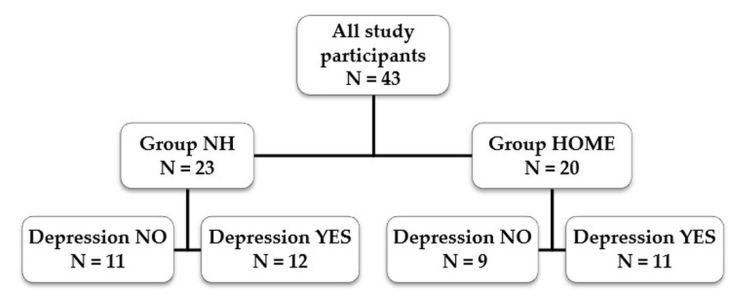
Study group—Geriatric Depression Scale (GDS) scores ≤ 9 points—Depression NO, score ≥ 10 points—Depression YES.

**Table 1 ijerph-18-09028-t001:** Characteristics of the studied groups.

	NH Group		HOME Group		t	*p*	Cohen’s d
	Mean	SD	Mean	SD			
N	23		20				
Age [years]	81.61	3.76	82.10	4.20	−0.40	0.6880	0.12
Weight [kg]	63.35	10.83	65.95	10.37	−0.80	0.4275	0.25

**Table 2 ijerph-18-09028-t002:** Physical activity and functional physical fitness of the subjects.

	NH Group			HOME Group					
	Mean	SD	±95% CI	Mean	SD	±95% CI	t/U	*p*	Cohen’s d
Step length [cm]	51.43	3.40	49.97–52.9	53.70	3.66	51.99–55.41	−2.11	0.0415 *	0.64
Steps/day [n]	1397.18	1071.77	933.71–1860.65	3583.54	2284.25	2514.48–4652.6	−3.92	0.0006 *	1.26
Distance/day [km]	0.76	0.56	0.51–1.0	2.09	1.21	1.52–2.65	−4.50	0.0001 *	1.45
Consumed Energy [cal]	28.83	21.15	19.68–37.97	93.32	58.98	65.71–120.93	−4.64	0.0001 *	1.50
Up and Go Test [s]	22.96	15.40	16.3–29.62	19.86	19.60	10.69–29.03	178.00	0.2098	0.39
Chair Stand Test [n]	8.39	5.28	6.11–10.67	11.75	6.17	8.86–14.64	144.00	0.0374 *	0.67

* *p* < 0.05.

**Table 3 ijerph-18-09028-t003:** Psychophysical self-assessment of the subjects.

	NH Group			HOME Group			t/U	*p*	Cohen’s d
	Mean	SD	±95% CI	Mean	SD	±95% CI			
GDS [points]	6.09	3.68	4.50–7.68	6.25	3.92	4.42–8.08	−0.14	0.8894	0.04
GARS [points]	33.57	10.79	28.9–38.23	27.35	11.77	21.84–32.86	133.50	0.0192 *	0.77
WHOQOL Total [points]	78.96	13.66	73.05–84.86	85.25	15.98	77.77–92.73	−1.39	0.1765	0.43
WHOQOL Health status [points]	5.87	1.49	5.23–6.51	6.40	2.14	5.40–7.40	−0.95	0.3585	0.291
WHOQOLD Phys. [points]	11.73	2.19	10.78–12.67	14.15	3.00	12.75–15.55	−3.05	0.0052 *	0.93
WHOQOLD Psych. [points]	12.78	2.38	11.76–13.81	12.69	2.32	11.6–13.77	0.13	0.8956	−0.04
WHOQOLD Soc. [points]	13.13	4.86	11.03–15.23	15.30	2.70	14.04–16.56	188.00	0.3036	0.32
WHOQOLD Envir. [points]	14.18	2.52	13.09–15.27	14.20	2.20	13.17–15.23	−0.03	0.9773	0.01
MMSE	24.17	2.87	22.93–25.42	25.15	3.48	23.52–26.78	−1.01	0.3269	0.31

* *p* < 0.05; GDS—Geriatric Depression Scale; GARS—Groningen Activity Restriction Scale; WHOQOLD—World Health Organization Quality of Life–Bref (total, health status and four domains: physical, psychological, social, and environmental); and MMSE—Mini-Mental State Examination.

**Table 4 ijerph-18-09028-t004:** Results of the physical activity, the functional physical fitness test and the psychophysical self-assessment questionnaires for subjects with and without depression.

	NH Group No Depression		NH Group Depression		t/U		HOME Group No Depression		HOME Group Depression		t/U	
	Mean	SD	Mean	SD	*p*	Cohen’s d	Mean	SD	Mean	SD	*p*	Cohen’s d
Step length [cm]	51.17	3.59	51.73	3.32	0.7021	0.16	54.56	3.05	53.00	4.10	0.3579	0.44
Steps/day [n]	1332.00	980.54	1468.28	1207.84	0.7684	0.13	4954.26	1976.14	2462.05	1927.48	0.0108 *	1.28
Distance/day [km]	0.75	0.54	0.77	0.61	0.9344	0.04	2.77	1.00	1.53	1.12	0.0188 *	1.17
Consumed Energy [cal]	29.52	20.64	28.07	22.67	0.8739	0.07	125.08	52.44	67.33	52.53	0.0249 *	1.10
Up and Go Test [s]	21.23	16.25	24.84	14.96	0.3099	0.23	10.48	2.17	27.53	24.13	0.0276 *	0.95
Chair Stand Test [n]	10.08	6.50	6.55	2.77	0.1962	0.70	12.44	2.51	11.18	8.16	0.3823	0.20
GDS [points]	3.67	2.50	8.73	2.87	0.0193 *	1.89	2.67	1.66	9.18	2.44	0.0044 *	3.06
GARS [points]	28.83	9.41	38.73	10.12	0.0164 *	1.02	20.22	2.82	33.18	13.18	0.0185 *	1.30
WHOQOL Total [points]	85.58	11.52	71.73	12.43	0.0089 *	1.16	94.56	15.53	77.64	12.26	0.0014 *	1.23
WHOQOL Health status [points]	6.67	0.89	5.00	1.55	0.0151 *	1.34	8.11	1.54	5.00	1.41	0.0011 *	2.12
WHOQOLD Phys. [points]	12.67	2.47	10.70	1.27	0.0489 *	0.99	16.38	2.02	12.32	2.38	0.1024	1.82
WHOQOLD Psych. [points]	13.78	1.27	11.69	2.85	0.0312 *	0.96	13.69	1.25	11.87	2.70	0.0627	0.84
WHOQOLD Soc. [points]	15.33	2.87	10.73	5.53	0.0089 *	1.06	16.67	3.00	14.18	1.89	0.0122 *	1.02
WHOQOLD Envir. [points]	15.54	1.67	12.69	2.50	0.8777	1.35	15.60	1.71	13.05	1.91	0.4704	1.40
MMSE	24.33	2.57	24.00	3.29	0.0193 *	0.11	26.11	2.52	24.36	4.06	0.0044 *	0.51

* *p* < 0.05; GDS—Geriatric Depression Scale; GARS—Groningen Activity Restriction Scale; WHOQOLD—World Health Organization Quality of Life–Bref (total, health status and four domains: physical, psychological, social, and environmental); and MMSE—Mini-Mental State Examination.

**Table 5 ijerph-18-09028-t005:** Two-way ANOVA analysis of variance taking into account the place of residence of the respondents and the presence of depression.

	Group NH vs. HOME	Depression Yes vs. No	Interaction Group and Depression	Cohen’s d
Step length [cm]	0.0391 *	0.6511	0.3384	0.303
Steps/day [n]	0.0000 *	0.0176 *	0.0086 *	0.866
Distance/day [km]	0.0000 *	0.0231 *	0.0192 *	0.765
Consumed Energy [cal]	0.0000 *	0.0179 *	0.0239 *	0.736
Up and Go Test [s]	0.4388	0.0519	0.1997	0.919
Chair Stand Test [n]	0.0515	0.1760	0.5174	0.938
GDS [points]	0.7177	0.0000 *	0.3374	0.304
GARS [points]	0.0194 *	0.0010 *	0.0004 *	1.696
WHOQOL Total [points]	0.0664	0.0004 *	0.6996	0.122
WHOQOL Health status [points]	0.0900	0.0000 *	0.0900	0.544
WHOQOLD Phys. [points]	0.0002 *	0.0000 *	0.1135	0.507
WHOQOLD Psych. [points]	0.9498	0.0055 *	0.8438	0.062
WHOQOLD Soc. [points]	0.3123	0.0041 *	0.0171 *	1.074
WHOQOLD Envir. [points]	0.7311	0.0001 *	0.8115	0.075
MMSE	0.2800	0.2937	0.4738	0.227

* *p* < 0.05; GDS—Geriatric Depression Scale; GARS—Groningen Activity Restriction Scale; WHOQOLD—World Health Organization Quality of Life–Bref (total, health status and four domains: physical, psychological, social, and environmental); and MMSE—Mini-Mental State Examination.

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
