# Peer review of "Assessment of the Psychophysical Sphere and Functional Status of Women Aged 75–90 Living Alone and in Nursing Homes"

_ijerph, 2021, doi:10.3390/ijerph18179028_

Round 1

Reviewer 1 Report

The subject of research is very relevant and interesting. However, I have a couple of comments that could improve your paper.
Firstly, you have some typos:

  • Line 20 and 436: the capital letter is missing: "to" "van".
  • Line 74: there is an opening parenthesis but the closing parenthesis is missing: "(the so-called".

Secondly, from lines 63 to 69 you talk about the aim of the investigation. It gets a bit repetitive, so you could put it together in one paragraph or try to change the writing so it doesn't seem like you're saying the same thing twice.

Finally, in the research methods section, you explain the period in which the research was carried out but not when the different physical and mental tests were done. I think it would be interesting if you could explain when the tests were done.  Several times a month, a week, at the beginning of the end of the research...? and why these times were chosen.

It is interesting work that I think you could continue with related analyses such as whether the depression or activity of women living at home is related to the previous profession of the participants.

Author Response

Dear Reviewer

Thank you very much for reviewing the work on behalf of all authors.

In response, I present the changes that we have made along with references to the comments:

- The typo indicated was corrected (line 20), but the prefix "van" was not corrected after verification. ". In the original article, "van" is written in lowercase in the authors' names (https://pubmed.ncbi.nlm.nih.gov/30849098/) (line 509).

- The purpose of the work was actually described twice. One version has been deleted (line 75).

- the tests were performed in the period from May to September 2019 in the morning (9:00 am-11:00am). In order to reduce physical and mental fatigue, tests were performed immediately after the completion of morning hygiene activities and after breakfast. Such information is included in the text of the paper (lines 99-102).

- we plan to continue the research by expanding the study group as soon as the epidemic situation related to COVID-19 allows. This will allow us to check other very interesting correlations between physical activity and the occurrence of depression and lifestyle, type of work, social status, etc.

Thanks again for your review.

Yours sincerely

Authors

Reviewer 2 Report

The authors present results from a study comparing differences in physical activity and mental states of elderly women living either in a nursing home of at their own homes. The topic of this study is timely and of relevance for an ageing society. However, there are a number of issues that have to be addressed to make this a stronger contribution.

There exists quite some research both in the medical field as well as in cognitive ageing research. The Introduction and Discussion should be expanded to give a more informative overview of relevant studies. Naturally, the authors would need to explain what contributes to the novelty of their research and define their research aims more specifically.

The main finding that people living at nursing homes are less physically active and less mentally healthy is quite plausible. However, the direction of causation cannot be inferred using cross-sectional data, as also pointed out by the authors.

The sample of 23 woman in nursing homes and 20 living at home is arguably small to derive meaningful inferences. I understand that persons between 75 and 90 years of age are a very special sample that may be difficult to recruit as participants. I think it is primarily this argument that speaks to possible publication of these results. Nevertheless, the authors should try to extend the sample in a possible revision, in particular, when they have access to elderly people who would be routinely tested with the reported assessment battery.

The authors test mean differences by means of t-test or U-test (in case of deviation from normal distribution; which I really liked to see). Nevertheless, I suggest testing results in form of ANOVAs. This way, the interaction of place of residence and depression can be tested statistically. In the current version, the authors just compare the presence or absence of mean differences, but they do not test if the difference of the difference is also significant. Effect sizes should be reported for all analyses.

Given the sparse data available, the authors should carefully inspect possible outliers (and possibly offer histograms of relevant variables in an appendix). Generally, both mean differences and relationships could be investigated by means of robust regression (e.g., using the MM estimator from the R-package MASS). Converging results would convince readers of the robustness of the findings. 

Minor issues (in chronological order)

- p. 1, Abstract: Colloquial contractions ("wasn't"; "weren't"; "didn't") should be avoided. 

- p. 2, l. 48: "Elderly people are often unable to fully take care of all their needs, which is the reason why they are often forced to live with caregivers, in nursing or retirement homes.": If this is know already, in what respect would this study yield novel insights?

- p. 2., l. 68: "The aim of the study was to assess ... women": Why was the aim limited to the study of women? Or, were fewer men willing to participate?

- p. 2, l. 84: "of the University School of Physical Education": Of which University?

- p. 2, l. 88: "The study groups did not differ significantly in terms of age (Table 1).": Possibly add "and weight" as this variable is also reported in Table 1. Actually, there are a number of more relevant variables that could contribute to group differences: educational level, life history of impairments, other illnesses, current social activities, contact with family and friends, hobbies and other activities maintained at old age. Possibly add such variable if available to describe the samples in a more informative way.

- p. 3, l. 91: "in the period from spring to autumn": Good to know, as current state of depression could be higher in winter. Anyway, please add year of data collection.

- p. 3, l. 106: "All the questionnaires were adapted to the Polish conditions.": This sounds like the authors translated themselves. Maybe rephrase as "Established Polish adaptations were used in this study."?

- p. 3 l. 110: "Values of 27-30 points are deemed to be the correct test result.": Maybe rephrase as "were deemed to indicate normal cognitive functioning"?

- p. 4, l. 143: "Daily physical activity was recorded for all the subjects, which in the HOME group turned out to be statistically significantly higher than in the NH group.": Split into 2 sentences.

- Table 2 / 3: Table reports M and SD, whereas Table 3 Median and IQR. If I understand the authors correctly, the reason was that they assume ordinal level for self-report scales? I think it is not unconventional to treat sum scores based on sufficient items as metric. Further, abbreviations of (sub-) scales should be spelled out in a note (relevant for all tables). 

- Tables 4 / 5 and corresponding text: I think reporting and interpreting correlations based on n=20/n=23 appears like a fragile snapshot. I suggest re-analyzing correlations (or robust regressions) for the entire sample to yield somewhat more reliable estimates. Following classical recommendations, the sample size should exceed n=60 for Pearson correlations.    

- Tables 6-8 and corresponding text: I suggest re-analyses using a 2x2 ANOVA with home (2) x depression (2) as factors. These analyses allow to test if the differences between depressed and non-depressed people differs between nursing-home vs. home as place of residence.

- p. 8, l. 239: "an attempt was made to understand problems connected with the place of residence": No specific factors concerning places of residence were tested – just nursing home vs. own home as global variables. 

- p. 9, l. 289: "It was shown that women with depression living in nursing homes did not differ in terms of daily physical activity and functional test results from women without depression in the same group. Subjects with depression who lived in their homes were less physically active and had worse functional capacity compared to women from the same group without depression.": This sounds like nursing homes can function to buffer negative effects of depression. In fact, when comparing all four subgroups, it turns out that non-depressive people at nursing homes are just as inactive as depressive people. Consider the variable "distance per day" for the four subgroups: nursing-home/depressive: .75; nursing-home/non-depressive: .77; own-home/depressive: 1.53; own-home/non-depressive: 2.77. These numbers suggest that even depressive people are more active when they live at their own homes. (Naturally, the direction of causation is unresolved.)

- p. 10, l. 356: "Subjects living alone performed a greater amount of daily physical activity, although the difference between the two groups was not significant.": I am not sure which variables or analyses the authors refer to. Consider again the variable "distance per day": nursing-homes: .76 vs. own-homes: 2.09; and the difference between both had a significance of p=.0001.

Author Response

Dear Reviewer

Thank you very much for reviewing the work on behalf of all authors.

In response, I present the changes that were made with references to the comments:

- As suggested in the review, the introduction and discussion were expanded. A literature review on the subject of the work has been added (Lines 51-58 and lines 330-332). References added: numbers 4 - 11.

- We plan to continue the research, increasing the study group as soon as the epidemic situation related to COVID-19 allows. Currently, access to care facilities is difficult and limited only to the procedures necessary for residents. Increasing the group of respondents will also allow checking other very interesting correlations between physical activity and the occurrence of depression and lifestyle, type of work, social status, etc.

- As suggested in the review, the intergroup comparisons were made using a two-way ANOVA, taking into account the place of residence of the respondents and the occurrence of depression. Effects size was calculated for each of the analyses (Tables 5 and 6) (lines 213-259). Table 4 was added to present descriptive statistics, including the presence or absence of depression in both study groups (lines 202-212). Thank you for the tip on how to analyse the results using regression analysis. We will definitely use it in further work

Minor issues (in chronological order) - replies

- "wasn't"; "weren't"; "didn't" replaced "was not"; "were not"; "did not" (lines 27, 28, 30, 34).

- The relationship between the physical and mental spheres of people of different ages is widely described in the literature. It is of particular importance in old age because the deterioration of the psycho-physical condition is influenced by involutional processes. Therefore, the assessment of physical abilities, matching the appropriate forms of activity and, above all, the promotion of an active lifestyle seems necessary and necessary among the elderly. According to the authors, this work fits in with this topic. (lines 51-58)

- The group of respondents was only women, because the authors, due to the restrictions related to COVID 19, had the possibility to conduct research only in female care homes. It was related to the consent or lack of consent to conduct research by the authorities of individual institutions. Of course, as soon as possible, we plan research in the group of men.

- Supplemented information on the Bioethics Committee "the Senate Committee on Ethics of Scientific Research of the University School of Physical Education in Wroclaw (Poland)" (lines 90)

- unfortunately, we currently do not have access to patient records. When conducting further research, we will certainly take into account variables related to the life history of the respondents.

- The tests were performed in the period from May to September 2019 in the morning hours (9.00-11.00 a.m.). In order to reduce physical and mental fatigue, tests were performed immediately after the completion of morning hygiene activities and after breakfast. Such information is included in the text of the paper (lines 99-102).

- Standardized adaptations of scales were used. The text has been changed as proposed (line 117).

- The text has been changed as suggested. The phrase "were deemed to indicate normal cognitive functioning" has been used (lines 121-122).

- The sentence was divided into two separate sentences: "Daily physical activity was recorded for all the subjects. In the HOME group turned out to be statistically significantly higher than in the NH group." (Lines 162-163)

- Table 2/3 - Descriptive statistics were standardized. Mean and SD values ​​were calculated for all analyzed parameters. In Tables 2 and 3, the p values ​​of the t or U test were left, while the effect size (Cohen's d coefficient value) was added. Explanations of abbreviations they contain have been added to the footers of all tables (lines 169, 182).

- Due to the small size of the group of respondents, the correlation analysis was abandoned (tables 4 and 5 and the corresponding fragment of the text were deleted)

- ANOVA was performed taking into account the interaction between the place of residence and the occurrence of depression. These results are presented in Tables 5 and 6 (lines 213-259). Table 4 was added, which presents descriptive statistics, taking into account the presence or not of depression in both study groups (lines 202-212).

- Other factors were not examined, therefore, as suggested, the sentence was changed to: "an attempt was made to determine the influence of the place of residence."(line 276).

- In order for the research results to be clearly described, the following sentence was added: "However, subjects with depression living in their own homes were characterized by greater physical activity compared to people living in nursing homes, regardless of their depression." (lines 330-331)

- Of course, the difference in physical activity between the two groups was significant. Thank you very much for pointing out a mistake that has been corrected (line 401).

Thanks again for your review.

Yours sincerely

Authors

Round 2

Reviewer 2 Report

The authors have quickly and adequately addressed most issues raised in my last review. Only few points should be considered in my eyes.

- The core limitation is the very moderate sample size, also when compared with other studies in ageing research. It should be emphasized that a larger sample is required to quantify effects more reliably. This would help the authors, too, if they plan to continue their research and collect more data.

- For consistency, effect sizes could be added also in Table 1, i.e., whenever a statistical test of group differences is reported. Possibly add a header line for all columns (r, p, d) referring to a test of group differences.

- Table 6 can be removed to save space, as it comprises only p values obtained in the ANOVAs. In fact, statistical significance is already indicated in Table 5 (by the asterisks), further, the effect size is directly given (hence, it does not need to be inferred from exact significance). 

Author Response

Dear Reviewer

On behalf of the authors, thank you very much for re-reviewing the work. In the submitted manuscript, we addressed any comments contained in the review.

- We agree with the remark that a relatively small study group is a significant limitation of this work. Relevant excerpt was added in chapter Research limitations (lines…).

- Table 1 was supplemented with a column containing the size of the effects

- As suggested in the review, the table containing the results of the post hoc test (Table 6) and references to it in the text has been removed

Once again, thank you very much for any comments that made the presented work more orderly and legible.

Yours sincerely

Authors